# Endocan in Acute Leukemia: Current Knowledge and Future Perspectives

**DOI:** 10.3390/biom12040492

**Published:** 2022-03-24

**Authors:** Håkon Reikvam, Kimberley Joanne Hatfield, Øystein Wendelbo, Roald Lindås, Philippe Lassalle, Øystein Bruserud

**Affiliations:** 1Department of Clinical Science, University of Bergen, 5020 Bergen, Norway; hakon.reikvam@uib.no; 2Department of Medicine, Haukeland University Hospital, 5021 Bergen, Norway; oystein.wendelbo@helse-bergen.no (Ø.W.); roald.lindas@helse-bergen.no (R.L.); 3Department of Transfusion Medicine and Immunology, Haukeland University Hospital, 5021 Bergen, Norway; kimberley.joanne.hatfield@helse-bergen.no; 4Inserm, Centre Hospitalier Universitaire de Lille, Institut Pasteur de Lille, U1019-UMR9017, University of Lille, 59000 Lille, France; plassalle@biothelis.fr; 5Center for Infection and Immunity, le Centre Nationale de la Recherche Scientifique, Univeristy of Lille, 59000 Lille, France; 6Centre d’Infection et d’Immunité de Lille, Equipe Immunité Pulmonaire, University of Lille, 59000 Lille, France

**Keywords:** endocan, p14 endocan fragment, protease, acute leukemia, chemotherapy, cytopenia, allogeneic stem cell transplantation, nonrelapse mortality, fluid overload

## Abstract

Endocan is a soluble dermatan sulfate proteoglycan expressed by endothelial cells and detected in serum/plasma. Its expression is increased in tumors/tumor vessels in several human malignancies, and high expression (high serum/plasma levels or tumor levels) has an adverse prognostic impact in several malignancies. The p14 endocan degradation product can also be detected in serum/plasma, but previous clinical studies as well as previously unpublished results presented in this review suggest that endocan and p14 endocan fragment levels reflect different biological characteristics, and the endocan levels seem to reflect the disease heterogeneity in acute leukemia better than the p14 fragment levels. Furthermore, decreased systemic endocan levels in previously immunocompetent sepsis patients are associated with later severe respiratory complications, but it is not known whether this is true also for immunocompromised acute leukemia patients. Finally, endocan is associated with increased early nonrelapse mortality in (acute leukemia) patients receiving allogeneic stem cell transplantation, and this adverse prognostic impact seems to be independent of the adverse impact of excessive fluid overload. Systemic endocan levels may also become important to predict cytokine release syndrome after immunotherapy/haploidentical transplantation, and in the long-term follow-up of acute leukemia survivors with regard to cardiovascular risk. Therapeutic targeting of endocan is now possible, and the possible role of endocan in acute leukemia should be further investigated to clarify whether the therapeutic strategy should also be considered.

## 1. Introduction

Endocan is a soluble dermatan sulfate proteoglycan that is expressed especially in endothelial cells but can also be detected in serum/plasma [1,2]. The serum endocan levels can be increased in cancer patients, including patients with untreated acute leukemia [3], but the systemic levels can be further modulated by anticancer therapy [3] and by inflammation. Local inflammatory reactions (i.e., a response to local tissue destruction) can be associated with the systemic acute phase reaction that is initiated and driven especially by tumor necrosis factor (TNF) α, Interleukin (IL) 1β and IL6, as well as other members of the IL6 cytokine family [4,5]. The acute phase reaction is characterized by altered levels of a wide range of serum proteins, but the overall acute phase protein profile varies and seems to depend both on the biological context and the cause of the inflammation [6,7]. Whether the effect of inflammation on systemic endocan levels shows a similar dependency on the biological context and/or the cause of inflammation is not known.

In this article, we review and discuss the possible biological importance of endocan in patients with acute leukemia, and the possible use of endocan as a prognostic biomarker and/or therapeutic target in acute leukemia patients. Most previous endocan studies in acute leukemia have included only or mainly patients with acute myeloid leukemia (AML). However, both AML and acute lymphoblastic leukemia (ALL) (i) are aggressive hematological malignancies characterized by bone marrow infiltration of immature myeloid or lymphoid blast cells, respectively [8,9]; (ii) are characterized by increased bone marrow angiogenesis and an adverse prognostic impact of extensive angiogenesis [10,11,12]; and (iii) can be treated with intensive and possibly curative conventional chemotherapy possibly combined with molecular targeted therapy (e.g., tyrosine kinase inhibitors) and/or allogeneic stem cell transplantation [13,14,15,16,17]. Finally, the acute promyelocytic leukemia (APL) variant of AML is characterized by specific genetic abnormalities, specific treatment and a better prognosis than other AML variants [18], and in the present article the term AML refers to the non-APL variants of the disease.

## 2. The Structure and Function of Endocan

The endocan gene consists of three exons, two introns and a 3′ end untranslated region [1]. Exon 1 and parts of exon 2 encode the 110 N-terminal amino acids; exon 2 also encodes the following phenylalanine-rich region as well as the last C-terminal amino acids (Figure 1). Genetic polymorphisms are very uncommon [2]. The complete endocan molecule consists of 165 amino acids and a dermatan sulfate chain including 32 disaccharide residues and being linked to the 137-position serine residue; the 113–118 positions include the phenylalanine-rich region [1,2]. The saccharide chain is regarded as critical for the biological functions of endocan, but the phenylalanine-rich region and especially the 116-position phenylalanine seem to be of particular importance for the function [1,2,19,20]. This is supported by studies of endocan-specific monoclonal antibodies in a subcutaneous tumor model; antibodies directed against the N-terminal parts inhibited tumor growth whereas antibodies targeting the C-terminal part of endocan had no growth-inhibitory effects [20]. Similar effects were also observed after alternative splicing of exon 2; the tumorigenic properties of endocan in an animal model were then lost [21]. Finally, the protein includes 18 cysteines that all are located in the 110 N-terminal amino acids of the protein [2].

Endocan is a 50 kDa soluble dermatan sulfate proteoglycan [2]. The dermatan sulfate chain can bind and thereby boost hepatocyte growth factor (HGF) induced intracellular signaling resulting in altered transcriptional regulation with induction of vascular endothelial growth factor (VEGF) and increased endothelial cell proliferation [1]. Endocan is also a stimulator of chemokine release, possibly through its effects on the transcription factor Nuclear factor kappa-light-chain-enhancer of activated B cells (NFκB) [22]. Furthermore, endocan can interact with the β2 integrin Lymphocyte function-associated antigen 1 (LFA-1 or CD11a/CD18), leading to an inhibition of the endothelial ligand Intercellular adhesion molecule 1 (ICAM1); this mechanism may represent a possible anti-inflammatory effect of endocan through inhibition of leukocyte extravasation/transmigration [2,23]. Finally, endocan can enhance the production of proinflammatory cytokines and the expression of adhesion molecules (Vascular cell adhesion molecule 1/VCAM-1) by endothelial cells [23].

The expression of endocan can be increased by Hypoxia-inducible factor 1α (HIF-1α) possibly via increased VEGF expression (i.e., endocan being both a target and a modulator of VEGF) [1], the proinflammatory cytokines TNFα, IL1β, Transforming growth factor β1 (TGFβ1) and Fibroblast growth factor (FGF) FGF2 [19], and the combination of HGF and VEGF [24]. The secretion is inhibited by Janus kinase 2 (JAK2)/Signal transducer and activator of transcription (STAT3) signaling (e.g., initiated by Interferon γ/(IFNγ) and Phosphoinositide-3-kinase (PI3K)-AKT serine-threonine kinase 1 (Akt) signaling (e.g., by Platelet-derived growth factor/PDGF, angiotensin, insulin) and lipopolysaccharide (probably through Toll-like receptor 4/TLR4 binding) [2,23]. Finally, primary AML cell/microvascular endothelial cell coculture experiments suggest that Angiopoietin 2 decreases local release of endocan [25], and this is probably a Tie-2 independent mechanism possibly mediated by α5/β1/β3 integrin chains [26]. Especially β3 integrins seem to be important for leukemogenesis/chemoresistance in human AML [27].

Endocan is produced by the vascular endothelium, especially by pulmonary and kidney endothelium [2]. It is continuously/constitutively synthetized and secreted; it does not belong to the endothelial glycocalyx and circulates freely in its glycosylated form at a concentration of about 1 ng/mL in healthy individuals [2]. These systemic levels of endocan are increased by a wide range of autoimmune and inflammatory disorders; it is also increased by cardiovascular diseases, diabetes, chronic kidney disorders, certain pulmonary diseases, and hypothyreosis [23]. Systemic endocan seems to have a half-life of one hour [28], and it is degraded by various proteinases including cathepsin G, neutrophil elastase, and proteinase 3 [28,29,30,31]. Cathepsin G seems to be the most important enzyme for degradation; it has a unique endocan degradation profile that includes 1–111, 1–115, and 1–116 endocan peptide fragments [30]. These fragments are referred to as p14 (14 kDa) endocan and they possibly exhibit a rigid structure due to their high number of disulfide bonds [2,30]. The p14 endocan can bind to β2 integrin LFA-1 and thereby inhibit the interaction between LFA-1 and endocan and restore the ICAM-1/LFA-1 interaction; the p14 fragment thus seems to have an antagonistic role compared to endocan [31].

## 3. Bone Marrow Expression of Endocan

Previous reports have shown that both endocan and the p14 endocan fragment can be detected in serum of healthy individuals [3,32]. The systemic levels of endocan are altered in several diseases, including both malignant disorders and autoimmune/inflammatory diseases [3,19,23,24]. However, these observations suggest that proinflammatory activity is only one out of several factors that can influence systemic endocan levels, and this seems to be true also for patients with acute leukemia as will be discussed in the following sections.

As described in a recent review, endocan is expressed mainly in lung and renal vasculature, but it is not restricted to endothelial expression and has been detected in normal serum as well as in several actively proliferative normal and malignant tissues [19,33,34]. A previous study described expression of endocan in AML and ALL blasts [35], but endocan did not reach quantifiable levels in previous proteomic studies of AML secretome, AML cell lysates, MSC secretome, MSC lysates, osteoblast secretome, or osteoblast lysates [36,37,38,39,40,41]. However, acute leukemia is characterized by increased angiogenesis, i.e., endothelial proliferation (see Section 4). We would therefore expect endocan to be released by endothelial cells in the bone marrow microenvironment, and this will possibly be the main bone marrow source.

A previous study investigated the release of endocan during coculture of microvascular endothelial cells and primary human AML cells [25]. Released endocan could then be detected during coculture for all investigated patients, but there was a four-fold difference between individual patients with regard to the endocan levels reached in the culture supernatants. These observations suggest that AML patients differ with regard to local endocan release in the bone marrow compartment, but it is not known whether this difference in local endocan release is associated with chemosensitivity or prognosis.

## 4. Systemic Endocan and p14 Endocan Fragment Levels in Untreated Acute Leukemia

### 4.1. Only Endocan but Not p14 Fragment Levels Show a General Increase but with a Wide Variation in Patients with Untreated Disease

A previous study described a significant increase in systemic endocan levels for AML patients compared with healthy controls [3]. There was also a wide variation between patients with regard to the serum endocan level [3], and as discussed in Section 3 this variation in systemic levels may reflect a difference in local endocan release from the bone marrow compartment [25,35]. Although our results presented in Appendix A have to be interpreted with care because they are based on observations in a relatively small but consecutive group of AML/ALL patients and without a validation group, these observations suggest that serum levels of the p14 endocan fragment for untreated acute leukemia patients do not show a similar general increase and wide variation as observed for endocan [3] (Appendix A; all p14 fragment results presented in this table have not been and will not be published elsewhere).

The p14 endocan fragment represents the main catabolite of endocan and was previously identified in the endocan degradation profile of cathepsin G [30]. Cathepsin G is expressed both by AML and ALL cells [42,43]. However, endocan can also be degraded by neutrophil elastase and proteinase 3 [30]; both these enzymes can be expressed by AML cells [44,45,46] and can be detected in plasma/serum [47,48]. In our opinion, it is not surprising that the systemic levels of endocan and p14 endocan fragment show no strong correlation in acute leukemia patients (Appendix A), because a similar observation with lack of correlation between these two endocan forms has also been described for sepsis patients [30] and their prognostic impact in intensive care patients with hematological malignancies also seems to differ [32]. Possible mechanisms for the discrepancy between systemic endocan and p14 endocan levels in acute leukemia patients could be altered or additional endocan degradation by other proteinases than cathepsin G [30,46], inhibition of cathepsin G activity by increased release of protease inhibitors (e.g., by the leukemic cells) [46], or altered cellular or extracellular matrix adhesion/binding of the p14 fragment. Modulation of protease activity is suggested as a possible explanation by the reduced activity of neutrophil elastase that has been described after intensive induction chemotherapy for acute leukemia; this reduction may thus be a chemotherapy effect, but it may also be a leukemia-associated effect that persists after remission induction [49]. To the best of our knowledge, it is not known whether similar reductions/modulations are present for proteases other than neutrophil elastase or whether this reduced activity can be induced by other types of intensive therapy or only for acute leukemia treatment.

To summarize, we have previously investigated endocan levels in acute leukemia patients at the time of diagnosis [3], and we have now investigated the p14 endocan fragment levels for a consecutive subset of these acute leukemia patients and for a group of healthy controls (Appendix A). Only the endocan levels seem to be generally increased in untreated acute leukemia patients, but it should be emphasized that there is a wide variation between individual patients and a subset of patients show endocan levels similar to the low levels observed in healthy controls [3]. In contrast, the p14 endocan fragment levels did not show a similar strong general increase or wide variation in untreated acute leukemia compared with healthy controls (Appendix A).

Based on the discussion above, we conclude that endocan should be further investigated as a biomarker to characterize the heterogeneity of acute leukemia patients at the time of diagnosis. However, in our opinion, the p14 endocan fragment should also be included in the future studies because it reflects biological characteristics other than endocan, and it may function as an endocan inhibitor or modulator [31]. The combined examination of both of these biomarkers may therefore help us to better understand the biological mechanisms behind the clinical impact of endocan.

### 4.2. Serum Endocan Levels Show a Wide Variation at the Time of Diagnosis for Many Human Malignancies and High Levels Are Often Associated with Adverse Prognosis

The possible role of endocan in carcinogenesis has been discussed in recent reviews [24,50]. Animal models suggest that endocan can stimulate carcinogenesis and promote cancer cell proliferation, suppress cancer cell apoptosis, and promote tumor angiogenesis [50]. Endocan overexpression has been associated with tumor progression and pharmacological suppression of endocan expression improved survival in a mouse model [1]. Finally, studies in subcutaneous xenograft models suggest that the phenyl-alanine-rich amino acid 113–118 region (see Figure 1) is important for stimulation of tumor growth [20].

A recent meta-analysis investigated the prognostic impact of endocan in human cancers [51]. They included both histochemical studies of local endocan expression and studies of serum/plasma endocan levels. Their overall results suggest that high endocan levels are an adverse prognostic factor both in gastrointestinal (i.e., gastric and colorectal cancer) and hepatocellular cancer, and this seems to be true both for systemic levels and local endocan expression. However, it should be emphasized that this prognostic impact was not observed in all available studies. Additional studies suggest that high endocan levels are associated with adverse prognosis also for pancreatic neuroendocrine tumors [52], non-small cell lung cancer [53,54], ovarian cancer [55], melanoma [56], and prostate cancer [57].

### 4.3. Angiogenesis and Cancer-Associated Inflammation; Common Characteristics of Acute Leukemias and Solid Tumors and Possible Mechanisms behind the Prognosictic Impact of Endocan in Cancers

Malignant diseases show biological similarities that are often referred to as hallmarks of cancer, and the observation of high endocan levels with an adverse prognostic impact in various malignancies suggest that the endocan levels are linked to or associated with such (a) hallmarks [51]. The possible prognostic impact of endocan expression in AML and ALL has not been investigated. However, the general increase and wide variation in serum endocan levels in acute leukemia at the time of diagnosis [3] together with the importance of angiogenesis for leukemogenesis/chemosensitivity and angiogenesis [10,11,12,15,58,59] suggest that future studies should investigate whether systemic endocan levels have a prognostic impact also in acute leukemias and not only in solid tumors. Endocan is now regarded as a possible therapeutic target in cancer treatment [60]. Furthermore, endocan can bind to LFA-1 and thereby inhibit its interactions with ICAM1/2 [2,23,31], and both primary AML [61,62] and ALL [63,64,65] cells can express LFA-1 as well as ICAMs. Thus, these expression data show that the molecular substrates for a direct functional effect of endocan on AML/ALL primary cells are present, and these observations further support our suggestion that the possible prognostic impact of endocan in human acute leukemia should be further investigated.

The acute leukemias are bone marrow disorders and endothelial, immunocompetent/inflammatory, and leukemic cells form an interactive network in the bone marrow microenvironment (Figure 2) [10,11,12,66,67,68,69,70,71]. Endocan is released by the endothelial cell and is both regulated by and is a regulator of inflammatory and leukemic cells. Angiogenesis is essential for tumor growth [66] and for the progression and even chemosensitivity of both AML and ALL, where extensive bone barrow angiogenesis is associated with an adverse prognosis [10,11,12,58,59]. Endothelial cells are also important for the formation the vascular stem cell niches that support both normal and leukemic hematopoiesis [10,11,12,67,71]. Thus, endothelial cell support and angiogenesis are common characteristics both for solid tumors and bone marrow infiltrating leukemias.

Another common characteristic of solid tumors and the bone marrow infiltrating acute leukemia seems to be the local infiltration of inflammatory cells in the microenvironment of the malignant cells; such local inflammation has been observed for several solid tumors and can then have a tumor-promoting effect and be associated with adverse prognosis [4]. Local bone marrow infiltration of immunocompetent/inflammatory cells that facilitates leukemic hematopoiesis (e.g., myeloid-derived suppressor cells/MDSC, various T cell subsets, NK cells) has also been described especially in AML [67,68,70,71], and, in particular, the local recruitment of macrophages with modulation of their polarization seems to be an important leukemia-supporting mechanism [68]. The local bone marrow hypoxia [69] and aging of myeloid cells [71] seems to further strengthen the leukemia-facilitating effects of immunocompetent/inflammatory cells.

Another common characteristic of solid tumors and acute leukemias seems to be induction of a systemic acute phase response; such systemic signs of inflammation and their association with adverse prognosis have been described for several solid tumors [4] and an adverse prognostic impact has also recently been described for human AML [72,73]. The possible association between the acute phase reaction and prognosis has not been investigated in ALL, but associations with adverse prognosis have been described for other lymphoid malignancies [74,75,76].

To conclude, endocan levels at the time of diagnosis show wide variations for patients with solid tumors and acute leukemia, and high endocan levels are associated with adverse prognosis for several tumors. The increased endocan levels in solid tumors probably reflect tumor-associated angiogenesis and/or inflammation, two processes that are fundamental in the development of both solid tumors and acute leukemias. Future clinical studies should, therefore, investigate whether the systemic endocan levels also have a prognostic impact in acute leukemia.

## 5. Systemic Levels of Endocan and the p14 Endocan Fragment in Acute Leukemia Patients with Severe Chemotherapy-Induced Bone Marrow Failure

### 5.1. Systemic Endocan Levels Are Decreased Following Intensive Chemotherapy but Increase as a Part of the Acute Phase Reaction during Febrile Neutropenia

We have previously investigated endocan levels in acute leukemia patients during the period of severe bone marrow failure due to intensive conventional antileukemic chemotherapy [3]. We have now, in addition, investigated the p14 endocan fragment levels for a consecutive subset of the acute leukemia patients included in these previous studies (Appendix A), and a comparison of endocan and p14 endocan fragment levels is thereby possible:Endocan levels are generally increased in untreated acute leukemia [3], whereas the p14 endocan fragment levels do not seem to show a similar general increase (see Section 4.1).The results presented in Appendix A showed low levels of the p14 endocan fragment in healthy controls and this is similar to previous studies [31,32]. Acute leukemia patients with chemotherapy-induced cytopenia also showed low levels of p14 endocan, even lower than the healthy controls (Appendix A). The p14 endocan levels did not increase even during febrile neutropenia due to bacterial infections (Appendix A), and this is also different from the endocan levels that increase during febrile neutropenia similar to the acute phase marker C-reactive protein [3].Several patients in complete hematological remission after intensive conventional chemotherapy still show increased endocan levels [77], whereas decreased p14 endocan fragment levels are observed even when patients have reached disease control, i.e., after induction of complete hematological remission (Appendix A).Previous studies could not detect any significant correlation between endocan and p14 endocan fragment levels in sepsis patients [30], and no strong correlation could be detected in our acute leukemia patients and healthy controls either (Appendix A).

A previous study investigated the endocan and the p14 endocan fragment levels in patients with hematological malignancies admitted to intensive care units [32]. These investigators concluded that endocan was a better predictor than the p14 fragment with regard to mortality and requirements for life-sustaining intensive care therapy (for details see Section 5.2). Thus, both this study [32] as well as previous studies in sepsis patients [16] suggest that endocan and p14 endocan fragment levels reflect different biological processes/characteristics; the results presented in Appendix A suggest that this is true also for patients with untreated acute leukemia and patients receiving intensive conventional antileukemic chemotherapy (i.e., therapy not including stem cell transplantation or new targeted therapies).

The treatment-related mortality for acute leukemia patients receiving conventional intensive therapy is low and few patients require transfer to an intensive care unit [13]. The patients included in our previous studies [3] (rather than patients transferred to intensive care units as described in [32], see Section 5.2) therefore represent the most common clinical situation for patients receiving intensive acute leukemia chemotherapy. Thus, the results from our previous study together with the results presented in Appendix A suggest that endocan is a better biomarker than the p14 endocan fragment for characterization of patient heterogeneity also for the majority of acute leukemia patients not requiring transfer to intensive care units for life-supportive therapy.

### 5.2. The Possible Prognostic Impact of Systemic Endocan Levels in Acute Leukemia Patients with Chemotherapy-Induced Bone Marrow Failure and Complicating Infections

Previous studies have shown that systemic endocan levels are low in patients with chemotherapy-induced neutropenia and no signs of complicating infections, the levels then increase during febrile neutropenia with increased CRP and then normalize when the signs of infection disappear during antibiotic treatment [3,78]. Thus, the systemic endocan level seems to increase as a part of the acute phase reaction during febrile neutropenia, both for children and adult patients. In contrast, our present p15 endocan fragment results (Appendix A) suggest that the serum levels of p14 endocan fragment remain low during the whole period of severe bone marrow failure, even during febrile neutropenia with increased CRP levels. Furthermore, only a minority of these patients showed a variation in p14 fragment levels corresponding to more than 20% of the first value when comparing before/during and during/after febrile neutropenia samples. Thus, in contrast to endocan, the p14 endocan fragment level does not seem to be a part of the acute phase reaction in acute leukemia patients with severe chemotherapy-induced cytopenia/bone marrow failure. This suggestion is also supported by a large clinical study describing a stronger association with survival for endocan than for the p14 endocan fragment for patients with hematological malignancies and life-threatening complications [32].

In a previous study, we described that the endocan serum levels increased during febrile neutropenia/bacterial infections compared with the levels immediately before the development of fever [3]. In contrast to the study discussed above [32], none of these patients developed life-threatening complications requiring transfer to an intensive care unit. It should also be emphasized that the preinfection endocan levels also showed a wide variation between individual patients, and this wide variation persisted during febrile neutropenia and even during clinical improvement after the infection, i.e., variations ranging from undetectable to exceptional patients with endocan levels exceeding 2.7 ng/mL. Taken together, these observations show that the acute phase reaction can modulate serum endocan levels (i.e., increased levels during infection). However, the maintained wide variation, even during febrile neutropenia, shows that other factors not reflected by the systemic CRP level have a major impact on systemic endocan levels during the period of severe bone marrow failure, even during fever/infection with development of an acute phase response with increased CRP levels.

Previous studies have suggested that decreasing endocan and possibly also p14 endocan fragment levels during sepsis and septic shock is an adverse prognostic factor associated with development of acute respiratory stress syndrome [30,79]. Even though our results presented in Appendix A should be interpreted with care because they are based on studies of a relatively small (although consecutive) group of AML/ALL patients and a validation group was not included, we would emphasize that our observations presented in Appendix A of neutropenic patients showed several similarities with previous studies in immunocompetent patients: (i) systemic endocan and p14 endocan fragment levels showed no strong correlation and (ii) the effect of febrile neutropenia on p14 fragment levels was generally weak (Appendix A). However, our present results (Appendix A) and previous studies [3] cannot answer the question whether decreased endocan/p14 endocan levels predict later development of severe (respiratory) complications because both of our studies are relatively small and none of our patients required transfer to the intensive care unit for respiratory or circulatory support.

A previous study compared the endocan and the p14 endocan fragment levels in patients with hematological malignancies admitted to intensive care units [32]. These patients were generally older (median age 60.5 versus 48 years) than the patients in our studies [3] (Appendix A); less than a third of them had neutropenia, only approximately one third had acute leukemia, and approximately one fourth had received chemotherapy. These investigators concluded that endocan was a better predictor than the p14 fragment with regard to mortality and requirement for life-sustaining intensive care therapy. Thus, both this study [32] as well as previous studies in sepsis patients [30] suggest that endocan and p14 endocan fragment levels reflect different biological processes/characteristics; the results presented in Appendix A suggest that this is true also for patients with untreated acute leukemia and patients receiving intensive conventional antileukemic chemotherapy (i.e., therapy not including stem cell transplantation or new targeted therapies).

The treatment-related mortality for acute leukemia patients receiving conventional intensive therapy is low and few patients require transfer to an intensive care unit [13]. The patients included in our previous studies [3] (rather than patients transferred to intensive care units as described in [32]) therefore represent the most common clinical situation for patients receiving intensive acute leukemia chemotherapy. Thus, the results from our previous study together with the results presented in Appendix A suggest that endocan is a better biomarker than the p14 endocan fragment for characterization of patient heterogeneity also for the majority of acute leukemia patients not requiring transfer to intensive care units for life-supportive therapy.

It is important to emphasize that the impact and effect of endothelial biomarkers may depend on the biological/clinical context. This is clearly illustrated by the systemic E-selectin (endothelial marker) and P-selectin (expressed by platelet and endothelial cells) levels that are increased when previously immunocompetent individuals develop meningococcal infections, whereas the systemic levels of these two markers decrease when acute leukemia patients with severe chemotherapy-induced pancytopenia develop complicating bacterial infections [80,81]. To the best of our knowledge, it is not known whether the impact of endocan as a predictive or prognostic biomarker can have a similar dependency in the biological context. However, the complex kinetics of endocan in sepsis patients [82] suggest that the biological context also has an effect on endocan.

Based on the discussion above, we conclude that endocan should be further investigated as a predictor of treatment-related toxicity/mortality in patients receiving intensive chemotherapy. However, in our opinion, the p14 endocan fragment should also be included in the future studies because it reflects different biological characteristics than endocan, and it may function as an endocan inhibitor or modulator [31]. The combined examination of both of these biomarkers (possibly also including other endothelial markers, see above) may help us to better understand the biological mechanisms behind the clinical impact of endocan in patients receiving antileukemic therapy.

## 6. Endocan and p14 Endocan Fragment Serum Levels in Acute Leukemia Patients Reaching Complete Hematological Remission after Intensive Chemotherapy

In our present p14 endocan fragment studies (Appendix A), we investigated the serum levels of endocan and the p14 endocan fragment for patients who had reached complete hematological remission. The patients showed a significant decrease in p14 endocan compared with healthy controls. In contrast, the endocan levels were still increased in the remission patients although this difference reached only borderline significance when compared with the controls; this is consistent with our previous observation for pretransplant endocan levels in allotransplant recipients [77]; all of these patients were in complete remission but the endocan levels varied and several patients had relatively high levels. Thus, the p14 fragment levels were still comparable to the low levels seen during chemotherapy-induced cytopenia/bone marrow failure, whereas for endocan there was a reduction but not a complete normalization compared with the increased levels observed for patients with untreated acute leukemia. This is an additional example that the association between total endocan and the p14 endocan fragment is weak or absent in many/most situations [30].

## 7. Endocan and Inflammation: General Mechanisms with a Possible Relevance for the Clinical Course after Allogeneic Stem Cell Transplantation

### 7.1. Modulation of Systemic Endocan Levels by Nonmalignant Diseases

As illustrated by the examples listed in Appendix A, endocan levels can be modulated by non-malignant diseases, including autoimmune disorders and infections, several malignancies, and anticancer treatment [3,19,23,24,50]. At least for certain in these diseases/patients, the endocan level seems to have a prognostic impact as discussed below.

Endocan expression can be increased by various proinflammatory cytokines, including IL1, IL6, and TNFα [1,19,23,24]. Some of these cytokines are also inducers of the acute phase reaction [4,5], but despite this overlap in their inducing cytokines there is no strong association between the systemic levels of endocan and the acute phase reaction. In our previous study, we observed increased systemic endocan levels during febrile neutropenia [3]. Despite this association of endocan with inflammation/CRP levels, there was a wide variation with regard to systemic endocan levels between individual patients with stable cytopenia and this wide variation persisted even after the induction of the acute phase reaction (i.e., increase in CRP levels) during febrile neutropenia [3]. Furthermore, decreased endocan levels have been described for sepsis patients that developed progression of inflammation to acute respiratory distress syndrome, and the most severe cytokine release syndrome during COVID-19 infection is associated with lower endocan levels than moderate inflammatory disease [83]. Finally, low pretransplant endocan levels are associated with increased posttransplant nonrelapse mortality, but for these patients, increased mortality is also associated with increased pretransplant CRP levels (discussed in Section 10 and Section 11). On the other hand, increased serum CRP as well as endocan levels in cancer patients are both associated with an adverse prognosis. Taken together, these observations suggest that inflammation is only one out of several factors that determine the systemic endocan level, and the final effect of inflammation on systemic endocan levels may therefore depend on the biological context as suggested in Section 6.

### 7.2. Endocan in the Long-Term Follow-Up of Acute Leukemia Patients—Systemic Endocan Level as a Sign of Endothelial Dysfunction and a Cardiovascular Risk Factor

Only one previous study has investigated endocan levels for patients in long-term complete hematological remission after intensive chemotherapy [84]. This study included 100 ALL survivors and they were tested at least 24 months after the last chemotherapy cycle; the patients had only received intensive chemotherapy and allotransplant recipients were excluded. These surviving patients showed significantly increased endocan serum levels. However, the increased levels for these patients probably reflect other biological mechanisms than the increased levels in patients with untreated disease or recently achieved complete hematological remission (see above), because the survivors also showed an altered lipid metabolism with increased total cholesterol, post-prandial glucose and triglycerides together with increased intima-media thickness, and also increased ferritin levels possibly reflecting an acute phase reaction and/or iron overload.

The carotid intima-media thickness is regarded as a marker for and a predictor of cardiovascular disease [85,86,87]. Although the study discussed above [84] showed that the carotid intima-media thickness was increased in long-term survivors after intensive conventional intensive chemotherapy for childhood ALL (i.e., pharmacological treatment not including stem cell transplantation or new targeted therapies), another study has questioned whether this parameter is a reliable marker for early vascular changes and cardiovascular risk in long-term survivors of childhood ALL [88].

Previous studies on biomarkers for cardiovascular risk in ALL survivors have mainly focused on biochemical parameters as predictors of cardiovascular events. The complexity of the cardiovascular risk evaluation for long-term survivors after treatment for childhood ALL is illustrated by two previous studies [89,90]. First, a cross-sectional analysis including 246 childhood ALL survivors (mean age 22.1 years; mean time since diagnosis 15.5 years) included analysis of CRP, TNFα, and IL6 together with several metabolic parameters [89]. Higher levels of TNFα were associated with obesity, whereas high CRP levels were associated with insulin resistance. The authors concluded that increased levels of proinflammatory biomarkers are associated with cardiometabolic risk factors/outcomes. Second, the PETALE cohort included 247 survivors (median age at inclusion 21.7 years) after childhood ALL [90]. The survivors showed increased frequency of hypertension/prehypertension, prediabetes, and dyslipidemia with high LDL-cholesterol. Thus, childhood ALL survivors have an increased frequency of cardiovascular risk factors, including metabolic syndrome, dyslipidemia, hypertension, and high LDL-cholesterol.

To summarize, induction of complete hematological remission is associated with a decrease in systemic endocan levels although the levels are still slightly increased compared with healthy controls. The reduction in endocan levels at remission compared with pretreatment levels is, in our opinion, probably due to disease control (see Section 6). In contrast, the persistent increase in endocan levels in long-term survivors is probably a part or consequence of the complex metabolic dysregulation that can be seen after treatment for childhood ALL. This dysregulation is also associated with altered levels of the proinflammatory markers CRP, TNFα, and IL6 (and possibly also ferritin) [84,89]. Several proinflammatory markers including both TNFα and IL6 can increase systemic endocan levels [1,19,23,24], and inflammation with increased systemic levels of proinflammatory markers (i.e., CRP, TNFα, IL6) is also associated with an increased risk of cardiovascular events in elderly individuals [91,92,93,94,95,96,97,98,99]. In our opinion, future systemic endocan levels should therefore be considered as a possible biomarker for cardiovascular risk in long-term survivors after treatment for childhood ALL and possibly also for other and adult patients who have received intensive chemotherapy for other (hematological) malignancies. Whether p14 endocan fragment levels can be used as a biomarker of cardiovascular risk cannot be judged based on the available reports. However, our present preliminary observations in acute leukemia patients in complete remission suggest that the p14 levels reflect other biological characteristics and not only mirror the endocan and CRP levels, and, in our opinion, may add information to the use of these two factors.

Previous studies have shown that immunocompetent patients with sepsis are heterogeneous with regard to variations in both endocan and p14 endocan serum levels during infection; adverse prognosis is possibly associated with decreased levels during infections [79]. This is in contrast to a previous study on intensive care unit patients with various hematological malignancies where high endocan levels at admission were associated with later increased mortality [32]. Because the early treatment-related mortality is very low after intensive antileukemic chemotherapy for relatively young patients (i.e., below 70–75 years of age) [13], additional and larger studies are needed to clarify whether variations in endocan or p14 endocan levels can predict later development of severe or fatal complications also for this group of patients.

### 7.3. Endocan Levels in Solid Organ Transplantation and in Cytokine Release Syndromes: Possible Biological Relevance for Allogeneic Stem Cell Transplant Recipients

Recognition of alloantigens is a common characteristic of solid organ transplant rejection and GVHD after allogeneic stem cell transplantation, i.e., recognition of major transplantation antigens in HLA-mismatched donors/recipients and minor antigens presented by matched HLA molecules [100,101,102,103,104]. Endothelial endocan expression has been investigated in patients with renal allografts and both the mRNA and protein levels increased in patients with graft rejection [105]. Similar observations have been made for patients with chronic renal allograft injury [106]. These observations suggest that donor endothelial cells with increased endocan release are involved in alloantigen-induced inflammation. This may be relevant for allogeneic stem cell transplantation even though the biological context of T cell reconstitution is different when host alloantigens are recognized by donor immunocompetent cells [107,108], because T effector cells seem to be essential in both cases and molecular mechanisms for antigen presentation/recognition will probably show similarities [100,101,102,103,104]. Endocan should, therefore, be further investigated as a biomarker of alloreactivity also in allogeneic stem cell transplantation.

Immunotherapy is now used in the treatment of acute leukemias [109,110], and a serious side effect of this therapeutic strategy is cytokine release syndrome [110,111,112,113]. This syndrome is especially seen in patients receiving haploidentical transplantation and is usually treated with anti-inflammatory/immunosuppressive therapy, i.e., IL1 or IL6 targeting therapy or systemic steroids [113,114,115,116]. A similar clinical picture referred to as the cytokine storm syndrome can also be seen in patients with COVID-19 infections, and systemic endocan levels then seem to predict the risk of additional severe respiratory complications [82]. In our opinion, future studies should clarify whether endocan can be used as a predictive/prognostic biomarker for severe cytokine-mediated reactions not only in COVID-19 infections but also in acute leukemia patients receiving immunotherapy or being at risk of severe immune-mediated toxicity (i.e., haploidentical transplantation, checkpoint inhibitor or CAR-T cell therapy, bispecific antibody treatment) that may require prophylactic, early systemic, or strengthened immunosuppression [113,117,118].

## 8. The Possible Prognostic Impact of Endocan in Allogeneic Stem Cell Transplantation

Microvascular or endothelial complications are well-known in patients receiving allogeneic stem cell transplantation [58,119]. Allotransplantation is used in the treatment of acute leukemia, and patients are usually transplanted after they have achieved complete hematological remission [120,121]. At this time point there is a wide variation in endocan levels between patients, and slightly increased systemic endocan levels may persist also after remission induction and during consolidation therapy (Appendix A) [77] whereas the levels of the potentially antagonistic p14 endocan fragment are low (Appendix A).

In a previous study, we investigated whether the pretransplant systemic endocan level was associated with survival after allogeneic stem cell transplantation [77]. The majority of these patients had AML or AML in complete hematological remission. This study focused on the early nonrelapse mortality; it was population-based and included 56 consecutive patients (48 with acute leukemia) receiving HLA-matched grafts from family donors. Most patients received myeloablative conditioning and graft versus host disease (GVHD) prophylaxis based on methotrexate plus cyclosporine A. There was a significant association between low preconditioning endocan levels and later steroid-requiring acute graft versus host disease (GVHD), especially liver and gastrointestinal GVHD. Furthermore, we used hierarchical cluster analysis to identify patient groups based on the preconditioning serum levels of the endothelial cell markers endocan, soluble E-selectin, and soluble VCAM-1. This analysis identified a minor patient subset that was characterized by low levels especially of endocan, but also of the two other markers, and this minority of 11 patients showed a significantly higher frequency of steroid-requiring acute GVHD and/or early multiorgan failure (8/11 versus 13/45, Fisher exact test, *p* = 0.013). Finally, we also compared the two main patient subsets identified in the cluster analysis, and the main cluster including the patients with the lowest preconditioning endocan levels showed a decreased survival due to increased early nonrelapse mortality. Taken together, these observations suggest that low pretreatment systemic endocan levels are associated with later severe respiratory complications; similar associations between low endocan levels and later development of severe complications have also been reported for sepsis patients [30,79] as well as COVID-19 patients [82]. However, one should emphasize that additional studies are needed to clarify whether endocan is associated with prognosis also when using unrelated matched donors, haploidentical donors, umbilical cord blood stem cells, alternative conditioning and especially reduced intensity conditioning, and other forms of GVHD prophylaxis/treatment.

A previous study on systemic pretransplant metabolic profiles also described an association with fluid overload that is a risk factor for acute GVHD and non-relapse mortality (see Section 11) [122]. Future clinical studies must clarify whether the prognostic association of endocan in allotransplant recipients [77] is secondary to and thereby reflects altered cytokine-mediated regulation of inflammation [123] and/or metabolism [122].

## 9. Endocan and Other Markers of Endothelial Dysfunction in Allogeneic Stem Cell Transplantation

Several vascular complications have been described for allotransplant recipients, including early posttransplant fluid overload or capillary leak syndrome, engraftment syndrome, transplant-associated microangiopathy, diffuse alveolar hemorrhage, and idiopathic pneumonia syndrome [124,125]. Several of these entities are relatively uncommon, and the possible role of endocan in their pathogenesis has not been investigated. For this reason, we will only discuss the possible role of endocan in the context of excessive fluid overload/retention.

The prognostic impact of early postconditioning fluid overload was first described by Tvedt et al. in 2015 [126]. A relatively small group of 100 allotransplant recipients was investigated, all patients received stem cell grafts from matched family donors, and the large majority of patients received myeloablative conditioning, peripheral blood stem cell grafts, and GVHD prophylaxis based on methotrexate plus cyclosporine. There was a strong correlation between the maximal weight gain (i.e., fluid overload) and the time until maximal weight gain was reached; patients with the largest weight increase had a higher preconditioning creatinine level, the weight gain showed no correlation with preconditioning albumin levels but was associated with lower albumin levels on days +14 and +28 posttransplant, and weight gain was not associated with cyclosporine levels. Furthermore, in adjusted analyses, weight gain and sibling-vs.-nonsibling family donor were associated with later acute GVHD, whereas weight gain, preconditioning CRP level, and preconditioning serum IL31 levels were associated with early nonrelapse mortality. The observations from this first study were later confirmed in several larger studies:Extensive fluid overload has a prognostic impact not only for patients receiving matched family donors but also for haploidentical stem cell grafts and for patients receiving HLA matched stem cell grafts in general [127]. Fluid overload seems to be more common in haploidentical transplantation [127].Fluid overload was also associated with increased nonrelapse mortality in patients allotransplanted with CD34^+^ enriched cells and GVHD prophylaxis without the use of calcineurin inhibitors [128].The adverse prognostic impact with increased nonrelapse mortality has also been observed in allotransplanted children [129].Concomitant early fluid overload is associated with increased mortality for pediatric allotransplant recipients developing severe respiratory failure in general [130] and children with severe respiratory failure due to engraftment syndrome [131].Severe fluid overload is also associated with nonrelapse mortality in cord blood transplantation [132].The Endothelial Activation and Stress Index (EASIX) is defined as lactate dehydrogenase (U/L) × creatinine (mg/dL)/platelets (10^9^ cells/L), and high EASIX at admission was a significant predictor of grade ≥2 fluid overload together with body weight below 80 kg in recipients older than 55 years, and with diabetes [133]. A high EASIX index also predicted severe posttransplant complications and transfer to intensive care units [134].

In these studies, excessive fluid overload was usually defined as at least grade 2 fluid retention, i.e., symptomatic fluid overload, with or without weight gain ≥10% to <20% from baseline, requiring ongoing diuretic therapy.

As described above, we previously examined pretransplant serum levels of the endothelial cell markers endocan, E-selectin, and VCAM [77], and endocan was the only single marker that showed a prognostic association, i.e., a significant association between low endocan levels and the development of severe posttransplant complications as well as increased posttransplant nonrelapse mortality. However, patients with serum endocan levels higher or lower than the median level did not differ with regard to postconditioning weight gain/fluid overload. To conclude, even though postconditioning fluid overload is an adverse prognostic factor and reflects endothelial dysfunction, endocan levels show no association with weight gain/fluid overload. The decreased endocan levels, thus, seem to have an additional prognostic impact and reflect other endothelial and/or vascular characteristics than the fluid overload.

A previous study reported an association between the preconditioning metabolic profile and the later degree of fluid overload in allotransplant recipients [122,126]. This study examined the serum levels of 766 metabolites for 80 consecutive allotransplant recipients. Patients with extensive postconditioning fluid retention showed increased pretreatment levels of metabolites associated with endothelial dysfunction (homocitrulline, adenosine), altered renal regulation of fluid and/or electrolyte balance (betaine, methoxytyramine, and taurine), and altered vascular function (cytidine, adenosine, and methoxytyramine). Additional bioinformatical analyses showed that excessive fluid overload was also associated with aminoglycosylation (possibly important for endothelial cell functions) and altered eicosanoid metabolism (also involved in immunoregulation and vascular regulation). These observations suggest the preconditioning metabolic profile is part of the predisposition for excessive posttransplant fluid overload and possibly also contributes to an adverse prognostic impact, whereas the adverse impact of low endocan levels probably reflects other biological mechanisms.

Taken together, these observations suggest that (i) the excessive fluid overload is caused by complex endothelial/cellular and metabolic mechanisms; and (ii) low systemic endocan levels reflect other risk factor(s)/mechanisms rather than the fluid overload.

## 10. Pharmacological Targeting of Endocan

Several pharmacological strategies have been suggested to target endocan; possible strategies to be considered in acute leukemia are summarized in Table 1 [1,3,135,136,137,138,139,140,141]. These strategies represent either endocan neutralization (e.g., monoclonal antibodies), targeting of the endocan release/level (e.g., statins, inhibition of proinflammatory cytokines), or targeting of endocan-induced modulation of intracellular signaling (e.g., VEGF, HGF, NFκB inhibition). It is not known whether any of these strategies will be effective and safe in the treatment of human acute leukemia, possibly with the exception of statins that have been investigated in several clinical AML studies. A combination of statins with conventional intensive therapy is safe, but results from randomized clinical trials are not available [142,143].

Additional studies are needed before therapeutic targeting of endocan is used in the treatment of human AML. As discussed in Section 4.3, high endocan levels are associated with adverse prognosis in several solid tumors, but despite similarities with regard to angiogenesis as well as cancer-associated inflammation and a wide variation in endocan levels for patients with acute leukemias, it is not known whether endocan levels have a similar adverse prognostic impact in human acute leukemia. If future clinical studies can demonstrate an adverse prognostic impact of high endocan levels at the time of diagnosis in AML and/or ALL, this would support further clinical studies of endocan-inhibitory therapeutic strategies (e.g., inhibition of endocan expression or downstream intracellular signaling (Table 1), antibody blocking of the N-terminal part [20]). On the other hand, low endocan levels seem to be associated with increased toxicity in allotransplant recipients [77]. A possible explanation for this observation is altered regulation of immunity/inflammation, because a similar prognostic impact of low endocan levels has also been described in certain immune-mediated diseases (Appendix A). However, future studies must clarify whether the decreased levels are important in the pathogenesis and thereby represent a driver of posttransplant immune-mediated complications through altered immunoregulation, or whether the decreased levels simply represent a secondary event that only reflects the severity of the disease, e.g., secondary to increased protease release and increased endocan degradation.

## 11. Summarizing Conclusions

Endocan is expressed by endothelial cells and increased cellular expression and/or systemic endocan levels are regarded as signs of endothelial dysfunction. Endocan expression is increased in many cancers and systemic levels are often altered in inflammatory disorders. Systemic endocan levels should therefore be further investigated as a possible prognostic biomarker in acute leukemia. First, increased bone marrow angiogenesis is an adverse prognostic parameter in acute leukemia [58,59] and at the same time systemic endocan levels are generally increased but at the same time show a wide variation in patients with untreated AML [3]. The possible association between endocan levels, bone marrow angiogenesis, and chemosensitivity/survival in acute leukemia should therefore be investigated further. It is not known whether endocan is associated with any of the identified prognostic parameters (i.e., a part of a high-risk phenotype) or whether it is an independent prognostic parameter in acute leukemia patients. Second, a large clinical study suggested that endocan levels can function as a prognostic marker in patients with hematological malignancies and life-threatening complications following therapy [32]; further studies must clarify whether endocan can be used as an early marker for severe complications following intensive antileukemic treatment. This should be investigated both for patients receiving conventional chemotherapy [3,32], new targeted therapies [135,136,137,138,139], and allogeneic stem cell transplantation [77,120,121,122,123,124,125]. Third, endocan may be useful in the long-term follow-up of relapse-free acute leukemia patients. Finally, pharmacological targeting of endocan is possible (Table 1), but this must be further investigated in preclinical models (including animal models) before clinical trials are considered. Several available studies suggest that endocan then is the most promising biomarker, whereas the p14 endocan degradation product seems less promising. However, despite this difference both these biomarkers should, in our opinion, be included in future clinical studies because they can be regarded as interacting markers and they seem to reflect different biological characteristics/processes.

## Figures and Tables

**Figure 1 biomolecules-12-00492-f001:**
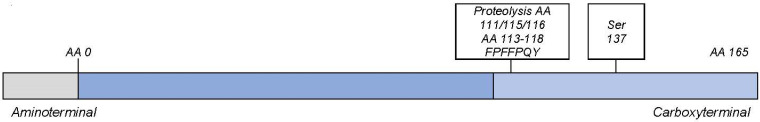
The structure of the endocan protein [3,4,5]. The mRNA encodes a propeptide including a signal peptide of 19 amino acids (grey color), whereas the complete secreted molecule includes 165 amino acids (dark blue plus light blue color). The molecule can be degraded by proteolytic cleavage, resulting in the p14 endocan fragments (dark blue color) that include 111, 115, or 116 amino acids. The phenylalanine-rich region is also indicated in the figure (amino acids 113–118). The dermatan sulfate chain is linked to Ser 137; this glycan chain has a molecular weight of 15–40 kDa.

**Figure 2 biomolecules-12-00492-f002:**
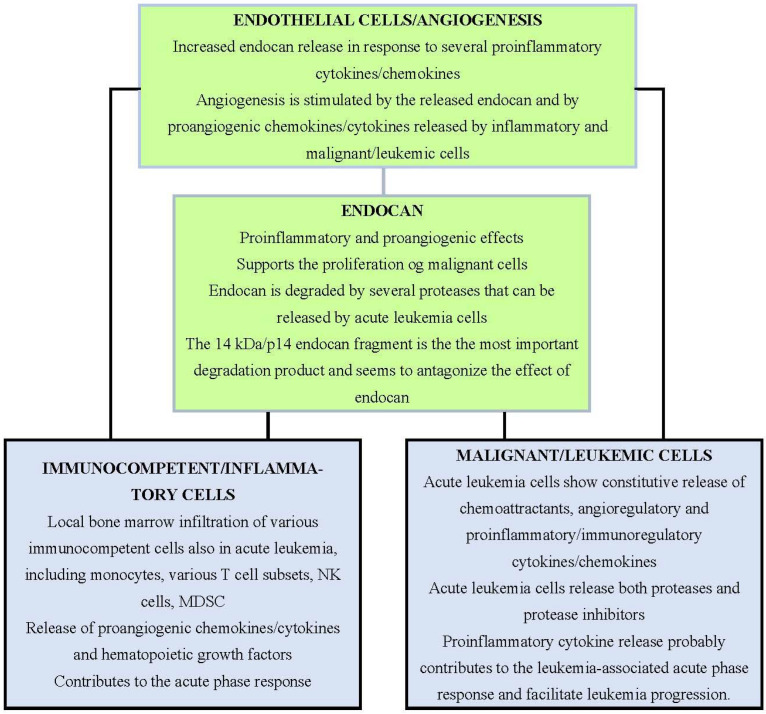
The bone marrow microenvironment in acute leukemia. Endothelial cells, immunocompetent cells, and the malignant/leukemic cells communicate through the extracellular release of soluble mediators and thereby form an interactive triangle. Endocan is released by endothelial cells; this release is modulated by, and also modulates, immunocompetent and inflammatory cells as well as leukemic cells through communication via the release of soluble mediators [2,4,10,11,12,19,20,21,23,31,42,43,44,45,46,50,58,59,67,68,69,70,71].

**Table 1 biomolecules-12-00492-t001:** Possible pharmacological strategies for targeting endocan and modulation of systemic endocan levels: a summary of strategies that may be relevant in AML [1,3,135,136,137,138,139,140,141].

Strategy	Description and Comment
ATRA [135]	A clinical study on AML patients with newly diagnosed leukemia showed that serum endocan levels increased after two days of oral ATRA monotherapy.
ATRA + valproic acid + theophylline [136]	Increased endocan serum levels were also seen in patients with newly diagnosed AML after seven days of combined treatment with ATRA plus the HDAC inhibitor valproic acid and theophylline.
Intensive induction acute leukemia therapy [3]	Patients with untreated AML have increased serum endocan levels; these levels decrease after intensive induction chemotherapy when patients develop severe pancytopenia, but slightly increased levels persist even after complete hematological remission is achieved.
Intensive consolidation acute leukemia chemotherapy [3]	Serum endocan levels also decrease during severe chemotherapy-induced pancytopenia following consolidation therapy.
Butyrate [137,138,139]	Butyrate is regarded as a histone deacetylase inhibitor and it has antileukemic effects in experimental studies of primary AML cells [135,136]; studies in colonic cancer cell lines have shown that butyrate increases endocan levels [138].
Inhibition of intracellular signaling [1,2,19,23,24,139]	Inhibition of TNFα, IL1, HGF, or VEGF initiated intracellular signaling would be expected to decrease endocan levels (see Section 2). Endocan promotes tumor cell proliferation in experimental studies through Akt-NFκB signaling; inhibition of either PI3K-Akt signaling or NFκB activity may therefore decrease endocan activity.
Statins [140,141]	Clinical studies suggest that statins can reduce systemic endocan levels.
Endocan-specific monoclonal antibodies [20]	Monoclonal antibodies directed against the amino-terminal parts of the endocan molecule seem to inhibit cancer-supportive effects of endocan in experimental studies (see Section 2).

## Data Availability

The data presented in this study are available on request from the corresponding author.

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
