# Peer review of "Endocan in Acute Leukemia: Current Knowledge and Future Perspectives"

_biomolecules, 2022, doi:10.3390/biom12040492_

Round 1

Reviewer 1 Report

I read the modified paper. No further questions

Reviewer 2 Report

The authors submitted a review paper aiming to discuss the use of endocan as a biomarker in human acute leukemias. The choice of the topic is justified. The paper, however, is extremely chaotic and its readability is rather low and should be improved. Extensive editing and well thought-through changes in logical sequence of the information provided is essential. 

Major points:

  • there is huge amount of text that is irrelevant to the topic, e.g. chapter in solid organ transplant, inflammation or human cancers in general
  • it is unclear if the paper is really a review or a short communication research paper (data in suplement); it is unclear why the authors provide some research data in supplementary materials - if I understand correctly this is a review paper and it is rather unusual to provide such data in a review manuscript; why the data is not cited? Wasn't it published before? If not - why and why are the authors trying to publish it in the review? If yes - is it ethically/legally acceptable to publish it again? The authors should decide what are they trying to publish - some preliminary data plus review of literature (kind of case series) or a real review of the literature

Minor points: 

  • the title of the paper is rather generic and not necessarily reflect the content. Please rephrase (i.e. reflect the main aim, which is clinical significance/use of endocan as a biomarker, the paper does not refer to chronic leukemias, so "acute leukemias" would be more accurate than "leukemias")
  • section titles do not reflect the content of the section, please rephrase
  • introduction consists of three completely separate parts (endocan, inflammation, leukemia) provided in illogical sequence making the readability of this fragment rather poor. I do not see much sense in providing separate definition of inflammation or some very rudimentary info on leukemias, as these information are not directly relevant to the paper and do not improve readability. Instead, the introduction start to resemble a glossary instead of logical and well thought-through introduction to scientific discussion that should appear in the next parts of the review. Please rephrase and improve.
  • the use of subjective language should be avoided (e.g. "It is not surprising that" verse 181)
  • some additional visual materials would be of great benefit for the paper (i.e. figures)
  • chapter 12 (pharmacological targeting of endocan) should be extended and provide justification to use endocan as a therapeutic target as it is unclear from the text what is the aim of such a treatment

Reviewer 3 Report

The Authors should at least mention limitations of their preliminary results on their small cohort of subjects indicanting that acute leukemias are referring both to myeloid and lymphoblastic types; therefore, their results should be carefully read based on the heterogeneity of their cohort.

Author Response

Please see the enclosed coverletter.

Round 2

Reviewer 3 Report

The Authors have addressed the raised questions; however, please clearly state that there are both myeloid and lymphoblastic leukemias in your study (Table S1) because these are completely different clinical entities and cannot be included only under the term of "acute leukemias". All other changes are good.

Author Response

Please see enclosed covering letter.

This manuscript is a resubmission of an earlier submission. The following is a list of the peer review reports and author responses from that submission.

Round 1

Reviewer 1 Report

In this study, Reikvam et al. have investigated circulating serum levels of total endocan and its degradation product, p14, in acute leukemias (both myeloid and lymphoblastic). Moreover, the Authors have reviewed available literature on this argument. However, several concerns need to addressed.

  1. This study has been conducted in a very small and heterogeneous population to draw any conclusion on clinical significance of circulating p14 levels in leukemias. A validation cohort is required.
  2. Supplementary Table 1 should be included as a main table.
  3. No information on the healthy control cohort is reported.
  4. Data are reported as tables, while it would be better to show them as bar graphs to visualize distribution and standard deviations.
  5. For some comparisons, statistical analysis used was not the best employed, and for correlation analysis, p values are missing.
  6. The Authors should describe in more details how patients with febrile neutropenia have been treated, as p14 levels might be affected not only by the type of disease, but also the type of chemotherapy used for induction, and the type of antibiotic treatment employed to treat the infectious complication.
  7. Association with CRP levels is reductive as this marker is not the only biomarker of febrile neutropenia.
  8. How the Authors translate these results into the next clinical step is not clear. Hence, the supporting data for their conclusion is insufficient.
  9. Finally, these data are inserted within a literature review on arguments that are far from presented results (even in allogeneic stem cell transplantation). Therefore, the Authors should consider to switch to a review article or a brief report.

Author Response

Please see the enclosed cover letter, the two sections General Response to editor and reviewers and Response to reviewer 1.

Reviewer 2 Report

Among molecules able to induce a “favorable” microenvironment in cancer, endocan seems to have a pivotal role.

In the paper the authors investigated serum level of endocan and of its degradation fragment p14, in a small series of acute leukemia at diagnosis and after intensive conventional therapy.

The paper I potentially very interesting but there are many weaknesses

  1. Numbers are to small and this can impact on statistical significance. Sample size must be increased
  2. Sample storage length and condition can negatively influence the results.
  3. The studied population is heterogeneous; AML and ALL are quite different and for origin and pathogenesis: the study should include only AML patients.
  4. Actually also AML are heterogeneous disease at clinical and molecular level: did they found any relationship with a particular leukemia subtype?
  5. There was any relationship with other clinical characteristics at disease onset?
  6. All patients received “conventional” chemotherapy:  all the same?
  7. Discussion is detailed and updated, but disjoined from experimental data.

Author Response

Please see the enclosed cover letter, the two sections General Response to editor and reviewers and Response to reviewer 2.

Reviewer 3 Report

The manuscript entitled “Endocan, inflammation and acute leukemia; studies of the p14 endocan fragment serum levels, review of endocan studies in acute leukemia and discussion of the possible importance of endocan in allogeneic stem cell transplantation” explores the role of serum level of the main endocan degradation product, named p14, in acute leukemia patients. The manuscript's topic is interesting and in the scope of the Biomolecules journal. However, the manuscript is confusing, unfocused, and seems to mix an original paper and a review. In order to be considered for publication, this manuscript needs a complete revision.

Specific comments:

  • The title is too long and not attractive enough.
  • The Introduction section needs to be improved. For instance, some information is excessive and unnecessary and needs to be better integrated and correlated.
  • The demographic data of healthy controls are missing. This information is crucial to evaluate if data between these two groups is comparable (Are the controls matched with cases?).
  • The statistical analysis description is missing.
  • Table 1 is only cited in the Discussion section.
  • In line 238, it is said that “11 AML patients”. Please, revise.
  • The title of Table 2 is too long, and some of the information should be moved to the footnote.
  • The results description is also confused. The results section should be clear, specific, and brief.
  • The Discussion section is also too long and disconnected. The Authors need to discuss the significance of their findings and future implications.
  • The Author should also include the study limitations.

Author Response

Please see the enclosed cover letter, the two sections General Response to editor and reviewers and Response to reviewer 3.

Round 2

Reviewer 1 Report

We thank the Reviewer for changing to a review article instead of an original one; however, data are still reported as tables, while it would be better to show them as bar graphs to visualize distribution and standard deviations. At least one graph or table with data should be reported in the main text.

Reviewer 3 Report

The approach taken by the Authors improved their manuscript. The title is still too long. I suggest something like "Endocan, inflammation and acute leukemia - Possible importance in allogeneic stem cell transplantation". The comments raised were addressed.